# Identifying High-Risk Patients with Advanced Midface Cancer: Personalized Surgical and Reconstructive Approach for Radical Resection

**DOI:** 10.3390/cancers17040585

**Published:** 2025-02-08

**Authors:** Daniel Bula, Jakub Opyrchał, Dominik Walczak, Łukasz Krakowczyk, Adam Maciejewski

**Affiliations:** 1st Department of Oncological Surgery with a Subunit of Reconstructive and Plastic Surgery, Maria Sklodowska-Curie National Research Institute of Oncology, 44-100 Gliwice, Poland

**Keywords:** midface cancer, microvascular reconstructive surgery, taxonomic prognostic care clusters

## Abstract

The study investigates personalized surgical and reconstructive approaches for advanced midface cancers. Analyzing data from 119 patients treated over a decade identified correlations between clinical factors, resective defect characteristics, and the optimal choice of free flaps for reconstruction. Patients were grouped into four clusters based on risk factors. Cluster B showed the worst prognosis (89% recurrence risk), while cluster C had the most favorable outcomes (6% recurrence risk). The 5-year overall survival rate was 95%, and disease-free survival (DFS) was 77%. Larger and more complex defects (Cordeiro type IIIA and IV) correlated with lower DFS and higher recurrence risk. Our study concludes that reconstructive surgery remains an effective personalized approach for managing advanced midface cancers, highlighting its critical role in achieving functional and cosmetic outcomes.

## 1. Introduction

Midface cancers develop in one of the most complex anatomical and functional structures of the human body. Therefore, reconstruction of such a complex postoperative defect is not an easy challenge in the field of microvascular reconstructive surgery [1,2,3,4,5,6,7]. A few important goals, such as local tumor control, restoration of bone and soft tissue structures and their function, facial contours, and finally, the best cosmetic effect, are expected to be achieved. Brown et al. [1] and Cordeiro et al. [2,3,4] clearly defined four to six classes of the postresective defect; however, the choice of the vascularized free or chimeric flap used for the reconstruction remains individually personalized [2,7]. Generally, combined treatment modalities of different malignant tumors are realized using protocols based on the results of evidence-based trials, although these can raise some uncertainties and doubts. Midface cancers are a specific group of malignant tumors since they encompass various key anatomical and functional structures. Until recently, locally advanced midface cancers were usually qualified for palliative therapy only, with no chance of being cured. Reconstructive surgery and rapid progress in this field have offered an attractive alternative to palliation, which is a high chance of local tumor control and very good cosmetic and functional effects.

The majority of the published studies on oncologic reconstructive surgery are mainly focused on the technical aspects of reconstructions and the use of various free flaps and/or modifications, and the majority of them are case reports or include a relatively small number of patients. One of the most important questions is whether there is any correlation between the patient’s clinical characteristics, the parameters of resective defects, and the choice of an optimal free flap; alternatively, there may not be any interrelationships and rules, and therefore, the reconstructive therapy may be clearly individually personalized. The answer to this question is the major aim of the present study, in which sophisticated taxonomic and cluster statistical methods were used.

## 2. Materials and Methods

The study group consisted of 119 consecutive patients with locally advanced midface cancer, who underwent microvascular reconstructive surgery as a primary treatment, in a single institution, by the same team of surgeons, over 10 years (2010–2020). All patients signed formal consent for treatment. The characteristics are presented in Table 1. The majority of tumors were in Stage T3 or T4, with three or more anatomical structures involved.

### 2.1. Resection

The extent of resection of the midface defects depended on tumor size and variety of the involved anatomical structures, including a part of the surrounding bony elements and soft tissues. The Cordeiro scale was used for the classification of post-ablative defects. Additionally, using two dimensions of the tumor (A and B) determined based on serial computed tomography imaging studies, the primary tumor area was calculated using the formula $P=\frac{3.14*A*B}{4}$.

### 2.2. Reconstruction

It was a challenge to reconstruct these complex postresective midface defect structures and achieve optimal, cosmetic, and functional effects. Therefore, the choice of a proper and optimal vascularized free flap was individually personalized. In the analyzed group, one of eight different free flaps was used (Table 2).

### 2.3. Statistics

Actuarial overall and disease-free survivals were estimated using the Kaplan–Meier method [8], and to estimate the hazard ratio for recurrence risk, Cox’s logistic regression was used [9]. Since the analyzed data included various postresective defects and individually chosen free flaps, taxonomic and cluster analysis [10] were used. Based on the arithmetically determined effects, a taxonomic analysis of the studied set of completely described patients was performed using the Marczewski–Steinhaus taxonomic distance metric [11]. It allowed for the separation of types of operated patients. Then, for specific types of patients, an additional analysis of variance (ANOVA) was performed, in order to determine the differences between the risk of local recurrence for individual indicators [12]. Statistical calculations were performed in the R v. 2.10.0 platform. Based on similarities and dissimilarities between the objects, the taxonomic matrix was estimated, and four major clusters were selected. Each cluster characterized similar factors that differed in the intercluster comparison. Finally, the tree dendrogram was estimated.

## 3. Results

### 3.1. Oncological Characteristics of the Examined Patients

The age of the 119 patients ranged from 22 to 94 years (mean age 58 years). The male-to-female ratio was 1:1.2. The BMI (body mass index) was determined for each patient. The largest group consisted of overweight patients—60 (50%), with six (5%) of them being obese. The second largest group was the group of patients with a normal BMI—53 (45%). The most common was squamous cell carcinoma—65 (55%); other histological types of tumors were as follows: basal cell carcinoma—15 (13%), adenoid cystic carcinoma—13 (11%), mucoepidermoid carcinoma—11 (9%), polymorphous adenocarcinoma—9 (7.5%), sarcoma—4 (3%), mucosal melanoma—2 (1.5%). In the squamous cell carcinoma group, the most common histological grading was G2—34 (52%). Patients with squamous cell carcinoma of other grades included in the study occurred with the following frequencies: GI—18 (28%) and GIII—13 (20%). The stage of the tumor according to the TNM system is presented in Table 1. In the study group, there was a significant predominance (71%) of patients with tumors in the T3 and T4 stage. Due to the high advancement of the primary cancer site, it was not possible to determine the location of the primary cancer site ex-post in all cases. Therefore, four groups of cases were distinguished depending on the location of the tumor. Most often (35%), it was the maxillary sinus. Considering the number of affected anatomical structures, it turned out that, in more than half (53%) of the cases, the primary tumor was extensive and involved four or more anatomical structures. However, based on imaging studies and clinical examination immediately before surgery, the research team assumed that most of the tumors originated primarily from the alveolar mucosa and/or palatine process of the maxilla. Therefore, the selected TNM scale for intraoral tumors was chosen to stage the patients. In this scale, the T4 feature occurs when the tumor invades adjacent structures only (e.g., through cortical bone of the mandible or maxilla or involves the maxillary sinus or skin of the face).

In the study group, 46 cases (39%) had metastatic lymph nodes (N+). However, it should be mentioned that in the group of midface tumors originating from the mucosa of the alveolar/palatine processes, the frequency of lymph node metastases was much higher than in tumors originating from, for example, the maxillary sinus and approached the level known from lower face tumors (tongue, floor of the mouth) [13,14].

### 3.2. Correlation Between Cordeiro’s Classification and Prognosis of Patients with Midfacial Defects

The actuarial 5-year overall survival was 113 (95%), and the disease-free survival (DFS) was 92 (77%). Using Cordeiro’s scale, the resective defects were classified in 9% of cases as type I (10), 41% of cases as type II A (49), 44% as type III A/B (52), and 3% as both type II B (4) and type IV (4). There were 21 patients who underwent a Cordeiro IIIB resection, i.e., total maxillectomy with orbital exenteration (18%). It is worth emphasizing that due to the removal of the orbital content, this group of patients requires a completely different approach in terms of reconstructive techniques and psychological care after surgical treatment. The area of defects ranged from 0.45 cm^2^ to 86 cm^2^, and half of them (59) were at least 25 cm^2^ or more. In 101 cases (85%), the radical resection was histologically negative; it was doubtful in the remaining 18 cases. The 5 yr DFS (*p* < 0.0001) correlated significantly with the Cordeiro’s type of the postresective defect. For type I and IIA, the 5 yr DFS was about 31% higher than for the type IIIB (89% vs. 57%). The Cordeiro’s type IV cases did not even reach the 5-year follow-up (Figure 1). In turn, according to the Brown classification, the course of the 5-year DFS curves, unlike those marked according to the Cordeiro classification, did not fully correlate with the increasing extent of the postresection defect. Although, the difference in the 5-year DFS between grades 1 and 3B was greater than between grades I and IIIB (Cordeiro) and amounted to 37% (84% vs. 47%), for grade 3A, it was comparable to 2A. This means that in the case of the Brown classification, the 5-year DFS did not correlate strongly enough with the increasing extent of resection. This suggests that in the assessment of the effectiveness of reconstructive microsurgery of the midface, the Brown classification seems to have a much weaker predictive and prognostic power than the Cordeiro classification.

Table 2 presents the types of flaps used and the number of patients in whom they were used. In all patients, a single-stage reconstruction of the postresection defect was performed. The most commonly used flap was the radial forearm free flap (RFFF) with or without a bone fragment (bRFFF), which was used in 51 patients (43%). In 57 patients (48%), one of the single microvascular bone flaps was used, and in the remaining 62 patients (52%), one of the soft tissue flaps was used.

### 3.3. Risk Factors of Local Recurrence

Local recurrence occurred in 27 cases (23%), among which, 16 recurrences (13%) developed during the first 12 months of follow-up. Histologically doubtful resection was found as the highest risk factor for local recurrence (Table 3). The incidence of postoperative complications was generally low. Partial flap necrosis occurred in 1.5% and vessels occlusion within the anastomosis in 6.7% of cases. Excellent functional effect, which means the full ability to take meals orally (i.e., without the need to maintain a gastrostomy) and the ability to breathe physiologically, were achieved in 90% of cases, and excellent aesthetic effect, which means the patient was fully satisfied with his/her appearance after reconstructive surgery, was achieved in 80%.

### 3.4. Taxonomic Matrix Dendrogram of Patients and Creation of Clusters Related to the Risk of Local Recurrence

A taxonomic matrix dendrogram was designed based on the clinical and surgical characteristics of all cases (Figure 2). The four clusters’ characteristics are presented in Table 4. Figure 3 likely suggests that cluster B is more similar to cluster A, less similar to cluster C, and dissimilar to cluster D. The detailed characteristics (Table 4) clearly suggest the dissimilarity of cluster B, when compared with clusters A, C, and D. Therefore, cluster B can be considered as the most unfavorable one, since doubtful radicalism corresponded to the highest risk of the local recurrence. On the contrary, cluster C looks to be the most favorable, because it predicts the lowest risk of local recurrence. Considering six variables in each of the four clusters, estimation of the 4 × 4 similarity matrix showed that clusters C and D represented roughly a high similarity index, which allows drawing a connecting line between them (Figure 3). Considering clusters C and D as a single one, the next step was to estimate the next similarity index using a 3 × 3 matrix. Clusters C and A were more or less similar; however, the connection lane between them was longer than that between clusters C and D. Therefore, the shorter the horizontal lines, the more similar the objects were. The result of the analysis shows the highest similarity for clusters C and D, a lower similarity for clusters C and A, and a dissimilarity (almost all analyzed parameters) for the cluster B. It is worth emphasizing that the parameter indicated in Table 4 is the most common in a given group. This means that in each cluster, in addition to, for example, type IIa or IIIa resection, other types occurred at a lower frequency (I, IV, etc.).

The clinical parameter characteristics of the resective deficits were classified using Cordeiro’s system (Table 2). Although the choice of a free flap has usually been highly individualized, the correlation between resective and reconstructive phases should not be ignored. The tissue composition, Cordeiro’s type, and radicalism of resective defect correlate, in some way, with the choice of optimal flap for the reconstructive phase. However, the algorithm should be considered as a general guide and not as a definitive indication of a single flap as the best one to be chosen, because each choice has its merits and its demerits.

## 4. Discussion

The clinical experience gathered over the years has contributed significantly to the organization and categorization of postresection defects. The type, extent, and location of these defects determine the selection of an appropriate and optimal free flap during the reconstructive stage. In 2000, Cordeiro and Santamaria [2,3,4] proposed a six-level scale for classifying resection defects, where each level corresponds to specific resection patterns of the midface bone scaffold. Conversely, Brown et al. [1] introduced a classification system for maxillectomy defects, which considers both the vertical and horizontal dimensions of the resected bone–soft tissue block. Both systems are utilized individually in clinical practice. In our study, extensive bone and tissue defects (classes IIIA-IV according to Cordeiro) were observed in 47% of patients, a figure comparable to the 54% of defects classified as 3A-4B using Brown’s system. This finding suggests that both Cordeiro and Brown’s classifications are comparable in assessing the extent of resection defects. Cordeiro et al. [3,4], based on the analysis of reconstructions performed on 100 patients with midface cancer over 15 years by a single surgeon in one center, developed an algorithm for reconstructing four types of maxillectomy resection stages. This algorithm accounts for the extent of the resection of midface bone and soft tissue blocks. The percentage of extensive resections classified as type IIIA-IIIB in Cordeiro’s study was comparable to our findings. However, while Cordeiro’s classification accurately categorizes the extent of resected bone and tissue defects, it does not directly guide the selection of reconstructive flaps, as evidenced by the lack of correlation between the type of flap used and the extent of resection. A publication worth noting is the work of Sun et al., in which, based on 105 cases, they examined the correlation between the effect of the treatment of squamous cell carcinoma of the middle part of the face and the Brown classification. The results of this work show a high correlation between survival and the degree in the Brown classification. However, it should be remembered that the largest limitation of this publication is the use of only one maxillectomy scale, where in our work, we compare the correlation with the Brown and Cordeiro scales [15] Chang’s [7] attempt to develop an algorithm for optimal free flap selection has not gained widespread practical utility, as individual decisions on flap selection often vary among surgeons. A thorough literature review reveals that a significant number of publications focus on single-patient reconstructions using specific flaps (case reports) [16,17,18,19,20,21,22,23,24,25]. Studies analyzing larger patient groups remain scarce and often involve small sample sizes [26,27,28]. For instance, Emara et al. [28] reviewed 772 PubMed articles on reconstructive surgery, selecting 14 for analysis. Of these, six were single-case studies, and the remaining eight analyzed fewer than 25 reconstructions. Their review concluded that radial or fibular flaps were the most commonly used, although the limited number of studies raises questions about the validity of this conclusion. Furthermore, the authors acknowledged that subjective assessments and a lack of treatment efficacy evaluations undermine the reliability of their findings. Other studies present similarly conflicting recommendations. Andrades et al. [29], based on 24 cases, recommended the use of a radial flap with a bone fragment (bRFFF) for complex zygomaticomaxillary defect reconstruction. However, nearly half of their cases had a follow-up period of less than two months, casting doubt on the reliability of their findings. In contrast, Futran et al. [30] suggested the fibular flap (FFF) as the optimal choice for similar defects in a study involving 27 patients with midface cancer, with a significantly longer follow-up period of 6 months to 6 years. These differing recommendations highlight the individualized nature of free flap selection, as defects, even in the same anatomical region, vary in shape and extent. Carrillo et al.’s work [31], although it includes a large group of cases (109), only analyzes resection treatment without providing any details regarding reconstruction. The authors mention T staging and the absence of surgery in the treatment strategy as the main adverse prognostic factors in treating midface cancer. Among the numerous published studies, two stand out due to their large patient cohorts. Ishimaru et al. [6] conducted a retrospective analysis of 2846 patients with head and neck cancer, including 468 with midface cancer, using data from Japan’s national database (2000–2012). Eight types of free flaps, including four bone flaps, were assessed, with a detailed evaluation of clinical factors predictive of reconstructive failures (3.3%). Similarly, Zhang et al. [5] analyzed 4640 reconstructive surgeries performed in a single center in Shanghai over 34 years. This cohort included 406 midface tumor cases (8.7%), with ten types of free flaps used, the most common being the radial flap (56%). However, the use of the radial flap decreased by half in the study’s later years. While the overall survival rates improved over time (91.9% to 98.4%), the authors did not specify whether these figures represented 2-, 3-, or 5-year survival. Zhang’s study assessed vascular complications (4.8%), focusing solely on anastomotic areas. If benign or non-advanced tumors were excluded, complication rates in the midface region could more than double. Despite the large sample size, the evolving sophistication of reconstructive techniques over 34 years raises concerns about the reliability of comparing the initial and final years’ outcomes. Nonetheless, Zhang’s work underscores the importance of individualized free flap selection and evolving reconstructive methods. The complexity of maxillofacial reconstruction lies in balancing the functional restoration with the aesthetic outcomes. The midface region, given its critical role in facial symmetry and function, presents unique challenges. A key consideration is ensuring adequate vascular supply, which directly impacts flap survival and overall reconstructive success. Technological advancements, such as 3D printing and virtual surgical planning, have begun to play a transformative role in improving outcomes [23,32,33]. For example, patient-specific models allow for precise preoperative planning and customization of bone flaps, which can significantly reduce the operative time and improve the flap fit. With the development and further introduction of robotics into reconstructive microsurgery, more and more reconstructions in such challenging locations as the midface will be performed with the assistance of a surgical robot, both in terms of free-flap harvesting and in performing micro anastomosis [34,35,36]. Despite these advancements, significant challenges remain. One such challenge is the lack of standardized protocols for flap selection. While algorithms like Chang’s [7] provide a framework, their limited adaptability to individual patient needs restricts their utility. A multidisciplinary approach, involving surgeons, radiologists, oncologists, and rehabilitation specialists, is essential to address the diverse aspects of maxillofacial reconstruction comprehensively. This collaborative strategy ensures that functional, aesthetic, and oncological outcomes are optimized. The role of long-term follow-up cannot be overstated. Monitoring patients over extended periods provides invaluable insights into the durability and functionality of different reconstructive techniques. It also allows for the early identification and management of complications, thereby improving patient satisfaction and quality of life. However, as highlighted by studies with limited follow-up periods, there is a pressing need for more longitudinal research to establish evidence-based practices in reconstructive surgery. In conclusion, while existing classifications and algorithms provide valuable frameworks for categorizing resection defects and guiding flap selection, reconstructive surgery remains highly individualized. Future studies should prioritize larger well-structured cohorts with standardized follow-up to enhance the reliability and applicability of findings. Moreover, embracing technological advancements and fostering multidisciplinary collaboration will be pivotal in advancing the field of maxillofacial reconstruction, ultimately leading to improved patient outcomes.

Until now, no study has focused on the analysis of any correlation between the clinical factors, characteristics of the resective defects, and the optimal choice of free flap for reconstruction. The majority of the studies concentrate on the technical details of the treatment and survival rates. Our results reveal a strong correlation between the Cordeiro’s type of resective defect and the 5-year disease-free survival (Figure 1) Types I and IIA correlate with a high 88% 5-year DFS, which is significantly lower by about 31% for type IIIB, and in the case of type IV, 5-year DFS cannot be expected. The designed matrix dendrogram (Figure 2) of the four selected case clusters shows that cluster B (Table 4) predicts the worst prognosis. The doubtful radicalism of tumor resection was the strongest risk factor for local recurrence (Table 4), together with Cordeiro’s type IIIA of the resective defect and a tumor size of 8–18 cm^2^; cluster C should be considered as the most favorable (Table 4, Figure 3). Our results representing a relatively large group of midfacial cancers clearly document the correlation between the clinical parameters, radicalism, size of resective defects, and the reconstructive part of surgical treatment. The presented results can be compared with Cordeiro et al.’s studies [2,3,4] and suggest that, for patients with midfacial cancer, the optimal choice of the free flap remains to be individually selected by a reconstructive team of surgeons.

The results of the taxonomic analysis showed that despite the highly individualized reconstructive stages and many variable factors determining their personalized course, it is possible to distinguish incompatible but similar clusters of patients with similar prognostic factors. However, it should be emphasized that such a procedure is not “a priori” and is not decisive for the selection of the optimal flap in the reconstructive stage but is “post factum” at the stage of assessing the results of treatment and after an appropriately long follow-up time after treatment. On the other hand, the differentiation of cluster B2 (Figure 3) suggests that the questionable radicality of the resection stage, together with BMI > 25 kg/m^2^, a IIIA defect according to Cordeiro, and a tumor surface > 10–15 cm^2^, should be considered “a priori” as risk factors for local recurrence, before starting the reconstructive stage, in order to carry it out with due precision and caution. In the future, patients in whom these features are identified will need to be treated differently. This may involve appointing a more experienced surgeon to perform a resection–reconstructive procedure or even organizing special oncological consultations where patients who meet certain criteria will be discussed in detail by more than one specialist.

Undoubtedly, the greatest limitation of our work is the huge diversity of patients suffering from advanced cancers. The location of the primary lesion in many patients with such advanced disease is impossible to precisely determine. Additionally, over the last 10 years, many changes have occurred in the methods of reconstruction and adjuvant treatment of patients with midface cancers. This fact may affect the survival and results of reconstruction of patients included in the study at the beginning and the very end. It should be emphasized that our study included only one center. This is both an advantage and a disadvantage. On the one hand, it allows for the standardization of the treatment process of patients included in the study, but at the same time, it prevents the possibility of obtaining follow-up in patients who completed oncological treatment in another center. In the future, the data can be compared with the results of other centers where reconstructive microsurgery techniques are used, which will allow for an increase in the number of patients included in the study.

Reconstructive surgery is one of the most complex treatments of various, usually locally advanced, cancers. Some attempts have been made to propose an algorithm for this personally individualized procedure [37,38]. Our results seem to be more detailed because the use of the recommended free flaps not only depends on the Cordeiro’s type of resective defect but also their size and tissue composition including bone structure (or not) and their length. However, the authors strongly believe that our results do not allow us to propose our algorithm, and they are rather a useful general guideline. One of the important factors is that the radicalism of the resection part was missing from the previous algorithms. Radical resection is a clear term, but “doubtful”, used in our study needs to be explained. It means that even if the surgical margins are histologically negative, they might be too narrow because of various topographical and/or technical reasons. In such a case, the risk of local recurrence highly increases (HR = 21.6, Table 4); therefore, whichever flap would be selected and considered as the best one might be misleading, because the local tumor control will fail sooner or later (cluster B in Table 4). Thus, radical resection is undoubtedly a key criterion for effective reconstruction. The present study clearly shows that the taxonomic and cluster analyses are prognostically very useful because they allow selecting clusters of patients (cluster C and D in Table 4) with the best prognosis, even if the reconstructive part of therapy is individually personalized.

## 5. Conclusions

Reconstructive surgery for midface cancers remains one of the most intricate challenges in oncologic treatment due to the complex anatomy and functional significance of the midface. Our study underscores the importance of correlating the clinical parameters, resective defect characteristics, and reconstructive strategies to optimize the oncological outcomes. The results highlight the prognostic value of Cordeiro’s classification, demonstrating its strong correlation with the disease-free survival rate. Furthermore, the taxonomic and cluster analyses proved instrumental in identifying prognostic subgroups, with radical resection emerging as a pivotal factor for achieving favorable results. While the selection of free flaps must be individualized, the study offers valuable insights into tailoring reconstructive approaches to patient-specific needs. Future research should prioritize large well-structured cohorts and integrate emerging technologies to refine evidence-based guidelines and enhance surgical outcomes in this challenging domain.

## Figures and Tables

**Figure 1 cancers-17-00585-f001:**
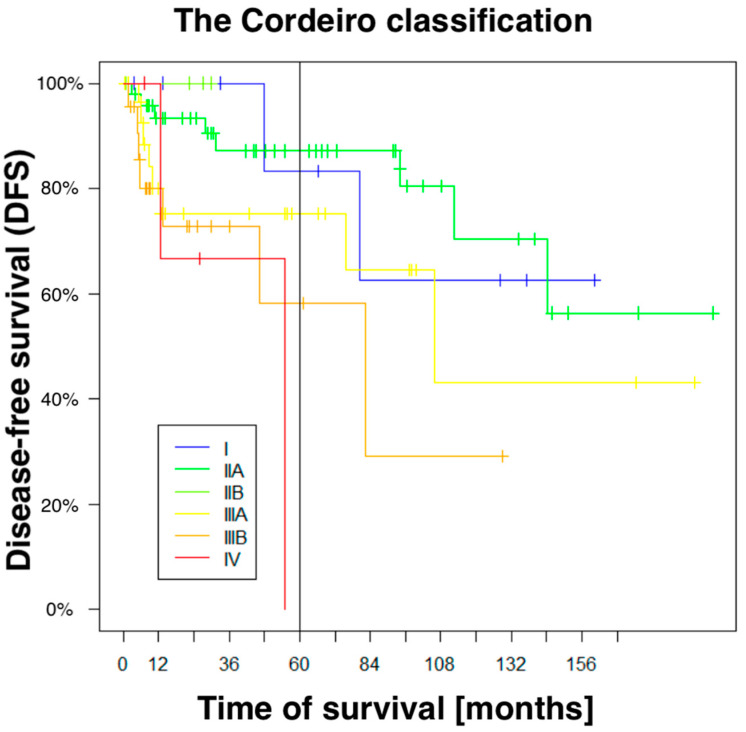
Disease-free survival (DFS) curves considering the Cordeiro classification. Since there were few patients with Cordeiro type IIB in the study group, it overlaps with Cordeiro type I.

**Figure 2 cancers-17-00585-f002:**
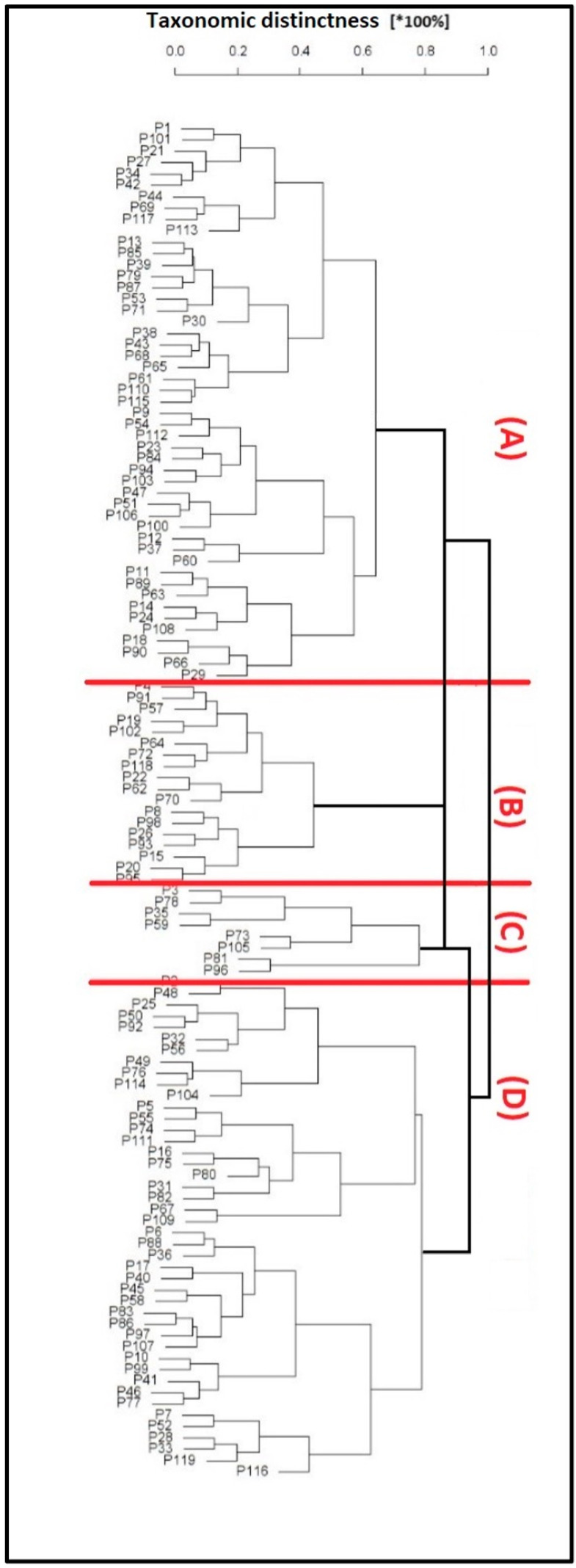
Taxonomic matrix dendrogram designed based on clinical and surgical characteristics of all cases.

**Figure 3 cancers-17-00585-f003:**
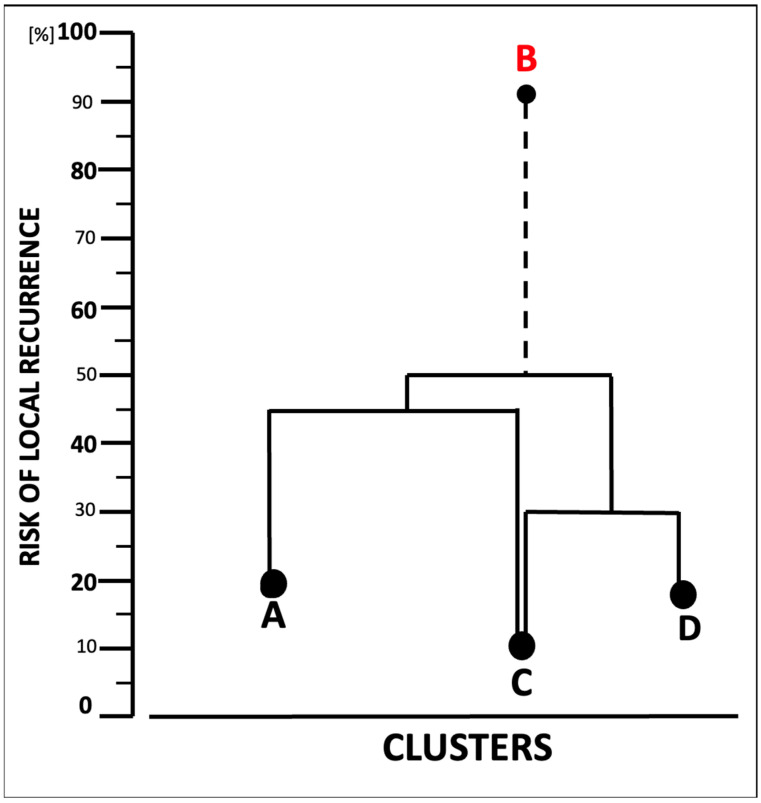
Taxonomic four clusters dendrogram (based on data from Table 4).

**Table 1 cancers-17-00585-t001:** Patient characteristics.

Parameters	Number of Cases—%
Age	
- Median	58 years
- Range	22–94 years
Sex Ratio (Male/Female)	1:1.2
Body Mass Index (BMI)	
- Normal (18.5–24.9 kg/m^2^)	45–53%
- Overweight	50–60%
- Obese	5–6%
Cancer Stage (TNM Scale)	
- T_1_ − T_2_ N_0_	18–21%
- T_1_ − T_2_ N+	11–13%
- T_3_ − T_4_ N_0_	43–51%
- T_3_ − T_4_ N+	28–34%
Tumor Site	
- Maxillary sinus	35–42%
- Hard palate	33–39%
- Maxilla	19–22%
- Skin of midface	13–16%
Number of Anatomical Structures Involved	
- 1	9–11%
- 2	16–19%
- 3	22–26%
- 4	20–24%
- >4	33–39%

**Table 2 cancers-17-00585-t002:** Types of free flaps used for reconstruction.

Type of the Flap	Number of Cases
**Microvascular bone flap**	
- Deep circumflex iliac artery bone flap (DCIAF)	22 (18%)
- Fibula free flap (FFF)	16 (14%)
- Bone radial forearm free flap (bRFFF)	15 (13%)
- Scapula bone free flap	4 (3%)
**Microvascular soft tissue flap**	
- Radial forearm free flap (RFFF)	36 (30%)
- Anterolateral thigh flap (ALTF)	20 (17%)
- Rectus abdominis muscle flap (RAM)	4 (3%)
- Facial artery musculomucosal flap (FAMM)	2 (2%)

**Table 3 cancers-17-00585-t003:** Hazard ratio of selected local recurrence risk factors.

Risk Factors	HR-Significance (*p*)
Cordeiro’s type of resection defect	1.5 (1.14–1.96)—0.004
Area of resected tissue block	1.78 (1.17–2.53)—0.006
Doubtful vs. radical resection	21.6 (9.1–51.3)—<0.0001
Recipient vessels:	
Cervical internal vs. external	0.38 (0.05–3.01)—0.36
Superficial temporal	2.07 (1.23–5.91)—0.013

**Table 4 cancers-17-00585-t004:** Characteristics of four clusters related to the risk of local recurrence.

Parameters	Clusters				
	Type A	Type B	Type C	Type D	Significance (*p*)
**No. cases**	49	18	8	44	-
**BMI index (kg/m^2^)**	27.5	25.2	20.4	21.9	0.0001
**Hb loss (g/dL)**	1.8	2.3	0.9	3.1	0.0001
**Tumor size (cm^2^)**	4–8	8–18	4–8	8–18	0.0016
**Cordeiro’s type of resective defect**	II A	III A	II A	III A	0.036
**Radical resection**	Yes	Doubtful	Yes	Yes	
**Risk of local recurrence**	12%	89%	6%	11%	0.0001

## Data Availability

Data is unavailable due to privacy and ethical restrictions.

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
