# Peer review of "Identifying High-Risk Patients with Advanced Midface Cancer: Personalized Surgical and Reconstructive Approach for Radical Resection"

_cancers, 2025, doi:10.3390/cancers17040585_

Round 1
Reviewer 1 Report
Comments and Suggestions for Authors
The authors used 119 locally advanced midface cancers to analyze the correlation between clinical factors, resection defect parameters, and free flap reconstruction methods. They found a strong correlation between Cordeiro type of each defect and 5-year disease-free survival, and identified four clusters by hierarchical clustering that showed different local recurrence risks. This may provide a new approach for studying midface defects due to surgical treatment. However, there are some points that should be reconsidered.
The interrelationships among the study's objectives, results, and conclusions are not always clearly stated and are difficult to understand. For example, the title reads “Personalized Surgical and Reconstructive Approach. However, “Personalized Surgical and Reconstructive Approach for Radical Resection” would be better.
The Introduction is insufficient to explain the purpose of this study. The lengthy discussion contains sentences that should be described in the Introduction; the “Discussion” on page 8, lines 156-163, and page 9, lines 257-259, could be moved to the Introduction section.
Statistics: Since the main part of this study consists of hierarchical clustering, the method should be described in more detail, including software, algorithm settings, and relevant papers available, rather than in a book (Ref. 10).
In their hierarchical clustering, the authors state that “the analyzed data included various post-resective defects and individually chosen free flaps (page 3, lines 72-73). However, the results depend on the variables selected. The list of variables used for hierarchical clustering must be clearly indicated. Are all data from Tables 1 and 2 included? Are data on neck dissection and surgical margin status included?
The paper should include section numbers. For example, “INTRODUCTION” and “MATERIALS AND METHODS” should be “1. Introduction” and “2. Materials and Methods”. To match the style of this journal, the results section can be divided into several subsections. For example, “Correlation between Cordeiro's classification and prognosis of patients with midfacial defects,” “Taxonomic matrix dendrogram of patients with midfacial defects and characteristics of the clusters obtained,” etc.
Why didn't the authors analyze the DFS for clusters (A-D)?
Discussion: The description of the use of free flaps and AI (page 8, lines 186-208 and page 9, lines 209-256) needs to be reduced. Instead, several additional papers describing the algorithm and prognosis could be added to discuss the results of this study.
For example,
Costa-H et al. Microsurgical reconstruction of the maxilla: Algorithm and concepts. J Plast Reconstr Aesthet Surg 2015, 68, e89.
Sun-Q et al. Does the Brown classification of maxillectomy defects have prognostic predication for patients with oral cavity squamous cell carcinoma involving the maxilla? Int J Oral Maxillofac Surg 2020, 49, 1135.
Carrillo-JF et al. Prognostic factors in maxillary sinus and nasal carcinoma. Eur J Surg Oncol 2005, 31, 1206.
More detail is needed on how the results and analysis of this study can contribute to the development of treatments for patients with midfacial cancer. Limitations also need to be explained.
Page 1, line 16: Stage T4 or T5 appears to be incorrect. Correction needed.
Table 1 and Results: The authors state that “ with 13% of them being obese (page 3, line 80). The percentage in Table 1 is 5%. This needs to be explained. In addition, the histologic diagnosis of each cancer should be described.
Page 3, lines 92-95: The number of patients needs to be shown, not just the percentage.
Figure 1: The IIB line is not clear. Furthermore, this figure includes Cordeiro's type I, but Table 4 does not include any type I cases. This discrepancy needs to be explained.
Table 3: The meaning of Temporal is unknown. Explanation needed.
Author Response
Comment 1 : The interrelationships among the study's objectives, results, and conclusions are not always clearly stated and are difficult to understand. For example, the title reads “Personalized Surgical and Reconstructive Approach. However, “Personalized Surgical and Reconstructive Approach for Radical Resection” would be better.
Response 1: The title was changed as reviewer suggested. New title clearly better describe the article
Comment 2: The Introduction is insufficient to explain the purpose of this study. The lengthy discussion contains sentences that should be described in the Introduction; the “Discussion” on page 8, lines 156-163, and page 9, lines 257-259, could be moved to the Introduction section.
Response 2: as recommended, the introduction was extended with suggested phrases from the discussion.
Comment 3: Statistics: Since the main part of this study consists of hierarchical clustering, the method should be described in more detail, including software, algorithm settings, and relevant papers available, rather than in a book (Ref. 10).
Response 3: The description of statistics has been described in more detail. Additionally, specific works presenting specific statistical tools have been cited in accordance with the recommendations. [8-12]
Comment 4: In their hierarchical clustering, the authors state that “the analyzed data included various post-resective defects and individually chosen free flaps (page 3, lines 72-73). However, the results depend on the variables selected. The list of variables used for hierarchical clustering must be clearly indicated. Are all data from Tables 1 and 2 included? Are data on neck dissection and surgical margin status included?
Respone 4: All variables were selected for hierarchical clustering. Unfortunately only those described in the table 4 had enough of taxonomic distinctness and were statistically significant. Data about neck dissection weren't statistically significant. We believe it was because in only 39% of cases, lymph node dissection was performed. In our experience, midface cancers are much less likely to require cervical lymph node dissection than, for example, lower face cancers. We believe that in the midface cancer, the N feature does not always represent the stage of the neoplastic disease as well as the size, extent of the tumor, and the possibility of its resection.
Comment 5 : The paper should include section numbers. For example, “INTRODUCTION” and “MATERIALS AND METHODS” should be “1. Introduction” and “2. Materials and Methods”. To match the style of this journal, the results section can be divided into several subsections. For example, “Correlation between Cordeiro's classification and prognosis of patients with midfacial defects,” “Taxonomic matrix dendrogram of patients with midfacial defects and characteristics of the clusters obtained,” etc.
Response 5 : Section number and subsection in results were added according to the recommendations
Comment 6: Why didn't the authors analyze the DFS for clusters (A-D)?
Response 6: In fact, the risk of local recurrence directly corresponds to DFS. In cases where local recurrence occurred, DFS decreased drastically. The risk of local recurrence was ultimately selected because it more directly corresponds to the effect of resection. Additionally, the main feature determining the different risk between clusters was questionable surgical margins, which is directly reflected in rapid local recurrence. DFS is influenced by more factors such as additional diseases and adjuvant therapy. These factors were beyond the scope of our study. In the future, we plan to analyze these factors, but due to the wide range of treatment dates in the patients, they will be of limited use.
Comment 7: Discussion: The description of the use of free flaps and AI (page 8, lines 186-208 and page 9, lines 209-256) needs to be reduced. Instead, several additional papers describing the algorithm and prognosis could be added to discuss the results of this study.
Response 7: Use of AI was completely deleted from the discussion. Additionally, the suggested articles have been added to the discussion. Thank you very much for bringing them to my attention.
Comment 8: More detail is needed on how the results and analysis of this study can contribute to the development of treatments for patients with midfacial cancer. Limitations also need to be explained.
Response 8: Both of these issues have been added to the discussion. Thank you for bringing them to our attention. They undoubtedly add to the substantive level of the discussion.Attempting to define a specific group of patients most at risk of complications was the most important goal of our research team. Introducing this knowledge into clinical practice is a top priority for us.
Comment 9: Page 1, line 16: Stage T4 or T5 appears to be incorrect. Correction needed.
Response 9: Corrected
Comment 10: Table 1 and Results: The authors state that “ with 13% of them being obese (page 3, line 80). The percentage in Table 1 is 5%. This needs to be explained. In addition, the histologic diagnosis of each cancer should be described.
Response 10: Percentage in results was corrected. Additionally we described all histological types of tumors and we added grading of SCC.
Comment 11: Page 3, lines 92-95: The number of patients needs to be shown, not just the percentage.
Response 11: Numbers of patients were added.
Comment 12: Figure 1: The IIB line is not clear. Furthermore, this figure includes Cordeiro's type I, but Table 4 does not include any type I cases. This discrepancy needs to be explained.
Response 12: Description of line IIB was added to the figure 1. Table 4 doesn't specify type I because the one in table was the most common in each cluster. It doesn't mean that other types did not occur.
Comment 13 : Table 3: The meaning of Temporal is unknown. Explanation needed.
Response 13: Temporal Superficial vessels description was added.
Reviewer 2 Report
Comments and Suggestions for Authors
Comments for authors
I read with great interest this article entitled "Identifying High-Risk Group of Patients with Advanced Mid-face Cancer: Personalized Surgical and Reconstructive Approach”. Resection and reconstruction of paranasal sinus tumors represent one of the most challenging procedures for head and neck surgeons. The manuscript is well done and the series is quite large.
There are some statements in the manuscript with which I completely agree: "...the choice of the vascularized free or chimeric flap used for the reconstruction remains individually personalized" (lines 38-39), and “… while existing classifications and algorithms provide valuable frameworks for categorizing resection defects and guiding flap selection, reconstructive surgery remains highly individualized”.
I have, however, a series of observations to make.
Although the primary aim of the study is to identify the best way to reconstruct the resection cavity, the authors also present the histology, the stage of the tumors, the radicalism of the resection, and the oncological outcomes. In fact, on lines 81-83, 85-86, 89-91, 92-93, and 95-97 they write: “The most common was squamous cell carcinoma (55%), while the remaining group of patients had a malignant tumor with a different histological diagnosis”, “.…a significant predominance (71%) were patients with tumors in the T3 and T4 stage”, “Considering number of affected anatomical structures, it turned out that in more than half (53%) of cases, the primary tumor was extensive and involved four or more anatomical structures”, “Actuarial 5-year overall survival was 95% and 77% of disease-free survival (DFS) respectively”, and “In 101cases (85%) radical resection were histologically negative, whereas it was doubtful in the 96 remaining 18 cases”.
Regarding the first statement, I think it would be appropriate to know the histology of 45% of patients with non-squamous cell carcinoma tumors. These data are important because the extent of resection and the resulting prognosis may vary for low-malignancy tumors (e.g., low-grade mucoepidermoid carcinoma).
Regarding the extension of the tumor and the structures involved (and therefore resected) I think that these data should be specified. In fact, a tumor is T3 when: "Tumor invades any of the following: bone of posterior wall of maxillary sinus, subcutaneous tissues, floor or medial wall of the orbit, pterygoid fossa, or ethmoid sinus”, and, excluding T4b, a tumor is T4a when: “Tumor invades any of the following: anterior orbital contents, skin of cheek, pterygoid plates, infratemporal fossa, cribriform plate, sphenoid or frontal sinuses”.
So, given that 71% of the cases were T3 or T4, in how many cases, for example, was the orbital floor resected and in how many cases was the orbital clearance performed? In how many cases was the medial pterygoid muscle resected, which is almost always infiltrated in a T4a? Furthermore, in my experience, resection of the medial pterygoid muscle always causes a more or less severe trismus. Regarding the aesthetic and functional outcome, the authors say: "Excellent functional effects 122 have been achieved in 90% of cases and esthetic effects in 80%" (lines 122-123), which seems too simplistic to me; I think that, above all, the functional aspect should be better specified.
As regards the rates of overall survival and disease free survival (95% and 77%, respectively) these seem astonishing to me. The vast majority of published studies report a 5-year DFS between 50% and 60% for T4 carcinoma. (Dulguerov P, et al. Nasal and paranasal sinus carcinoma: are we making progress? A series of 220 patients and a systematic review. Cancer. 2001 Dec 15;92(12):3012-29. doi: 10.1002/1097-0142(20011215)92:12<3012::aid-cncr10131>3.0.co;2-e, Turner JH, Reh DD. Incidence and survival in patients with sinonasal cancer: a historical analysis of population-based data. Head Neck. 2012 Jun;34(6):877-85. doi: 10.1002/hed.21830, Mirghani H, et al. Sinonasal cancer: Analysis of oncological failures in 156 consecutive cases. Head Neck. 2014 May;36(5):667-74. doi: 10.1002/hed.23356). In fact, the 2017 WHO Classification of Head and Neck Tumours states verbatim: ”The 5-year overall survival rate for squamous cell carcinoma is approximately 50-60%, and is stage-dependent”.
Author Response
Comment 1:
Although the primary aim of the study is to identify the best way to reconstruct the resection cavity, the authors also present the histology, the stage of the tumors, the radicalism of the resection, and the oncological outcomes. In fact, on lines 81-83, 85-86, 89-91, 92-93, and 95-97 they write: “The most common was squamous cell carcinoma (55%), while the remaining group of patients had a malignant tumor with a different histological diagnosis”, “.…a significant predominance (71%) were patients with tumors in the T3 and T4 stage”, “Considering number of affected anatomical structures, it turned out that in more than half (53%) of cases, the primary tumor was extensive and involved four or more anatomical structures”, “Actuarial 5-year overall survival was 95% and 77% of disease-free survival (DFS) respectively”, and “In 101cases (85%) radical resection were histologically negative, whereas it was doubtful in the 96 remaining 18 cases”.
Regarding the first statement, I think it would be appropriate to know the histology of 45% of patients with non-squamous cell carcinoma tumors. These data are important because the extent of resection and the resulting prognosis may vary for low-malignancy tumors (e.g., low-grade mucoepidermoid carcinoma).
Response 1: Thank you for pointing it out. I added the histology report of other non-scc cases. Additionally grading of SCC group was added.
Comment 2:
Regarding the extension of the tumor and the structures involved (and therefore resected) I think that these data should be specified. In fact, a tumor is T3 when: "Tumor invades any of the following: bone of posterior wall of maxillary sinus, subcutaneous tissues, floor or medial wall of the orbit, pterygoid fossa, or ethmoid sinus”, and, excluding T4b, a tumor is T4a when: “Tumor invades any of the following: anterior orbital contents, skin of cheek, pterygoid plates, infratemporal fossa, cribriform plate, sphenoid or frontal sinuses”.
So, given that 71% of the cases were T3 or T4, in how many cases, for example, was the orbital floor resected and in how many cases was the orbital clearance performed? In how many cases was the medial pterygoid muscle resected, which is almost always infiltrated in a T4a?
Response 2: That's a very good observation. I am very aware that grouping patients with t3 and t4 tumors may introduce ambiguity regarding the characteristics of the study group. Unfortunately, the biggest problem in studying the survival and treatment effects of patients with advanced midface cancer is the significant heterogeneity of the study group. In order to increase the possibility of obtaining statistically significant results, we have very reluctantly simplified the presentation of patients by grouping the T feature. On the other hand, in order to precisely analyze the impact of the tumor advancement and the extent of resection on treatment results, the research team decided to classify all patients according to the Brown and Cordeiro classification. The number of patients in the given classes explains in detail which anatomical elements were subjected to resection.
Comment 3 : . Regarding the aesthetic and functional outcome, the authors say: "Excellent functional effects 122 have been achieved in 90% of cases and esthetic effects in 80%" (lines 122-123), which seems too simplistic to me; I think that, above all, the functional aspect should be better specified.
Response 3: We are very grateful for your attention to this detail. The omission of a detailed description of the functional and aesthetic results of the reconstructive procedure was introduced into our manuscript after a long debate by the research team. This was done in order to shorten and simplify it. However, in view of this comment, a full explanation of the significance of the aesthetic and functional results obtained was added to the manuscript.
Comment 4 : As regards the rates of overall survival and disease free survival (95% and 77%, respectively) these seem astonishing to me. The vast majority of published studies report a 5-year DFS between 50% and 60% for T4 carcinoma. (Dulguerov P, et al. Nasal and paranasal sinus carcinoma: are we making progress? A series of 220 patients and a systematic review. Cancer. 2001 Dec 15;92(12):3012-29. doi: 10.1002/1097-0142(20011215)92:12<3012::aid-cncr10131>3.0.co;2-e, Turner JH, Reh DD. Incidence and survival in patients with sinonasal cancer: a historical analysis of population-based data. Head Neck. 2012 Jun;34(6):877-85. doi: 10.1002/hed.21830, Mirghani H, et al. Sinonasal cancer: Analysis of oncological failures in 156 consecutive cases. Head Neck. 2014 May;36(5):667-74. doi: 10.1002/hed.23356). In fact, the 2017 WHO Classification of Head and Neck Tumours states verbatim: ”The 5-year overall survival rate for squamous cell carcinoma is approximately 50-60%, and is stage-dependent”.
Response 4: The results obtained were also a surprise to us. Due to the fact that we are the National Institute of Oncology, we compare our patient treatment results to the latest publications and we are aware that they are very optimistic. However, our interpretations show that such a good result could have been achieved because, in addition to very advanced patients, the study included cases with low degrees of malignancy and low local advancement. In the future, we plan to include cases from other centers in our country in the study. This will undoubtedly increase the number of cases and allow for the unification of results with international reports.
Reviewer 3 Report
Comments and Suggestions for Authors
Parameters that were taken into account for the formation of clusters are not sufficient to allow correct conclusions. A mandatory parameter for assessing the risk of local recurrence is the histopathological analysis of the tumor. Also, the oncological treatment performed after the surgical intervention is important. The existence of concomitant diseases could be a factor influencing the risk of local recurrence.
The statistical analysis is correctly performed but could be improved by adding the mentioned parameters
Comments on the Quality of English LanguageEnglish writing should be improved
Author Response
Comment 1: Parameters that were taken into account for the formation of clusters are not sufficient to allow correct conclusions. A mandatory parameter for assessing the risk of local recurrence is the histopathological analysis of the tumor. Also, the oncological treatment performed after the surgical intervention is important. The existence of concomitant diseases could be a factor influencing the risk of local recurrence.
Response 1: Thank you for drawing attention to this problem. The comment is of course correct. Unfortunately, the clustering methodology has its limitations. Our data show that the main feature that determined the increased risk of local recurrence, among others presented in Table 4, were questionable surgical margins (radical resection status). These were mainly dependent on the stage of the tumor and its histological characteristics. Our goal was to examine the conditions (patient and resection-reconstruction procedure) in which the occurrence of local recurrence is most likely. Of course, we are aware that in the distant follow-up, factors such as adjuvant treatment, additional diseases and socio-economic status will play a significant role.
Comment 2 : The statistical analysis is correctly performed but could be improved by adding the mentioned parameters.
Response 2 : In addition, for additional explanation, a fragment of the methodology concerning statistics has been rewritten. In the future, we would like to expand our database to include the effects of treatment in other facilities. Then, a revised statistics will be conducted, including your observations. Thank you very much for this accurate observation.
Round 2
Reviewer 1 Report
Comments and Suggestions for Authors
The authors respond appropriately to most points raised by the reviewer.
However, the authors stated that “Table 4 doesn't specify type I because the one in table was the most common in each cluster. It doesn't mean that other types did not occur.”
This needs to be described in the text.
Author Response
Comment 1:
However, the authors stated that “Table 4 doesn't specify type I because the one in table was the most common in each cluster. It doesn't mean that other types did not occur.”
This needs to be described in the text.
Response 1: Thank you very much for your insightful review. I think that all these changes have greatly increased the value of the manuscript. An explanation of the value of Table 4 in lines 184-187 has been added.
Reviewer 2 Report
Comments and Suggestions for Authors
I have carefully read the authors' responses to my observations. The additions made regarding the histology of non-squamous cell carcinoma cases are now exhaustive. However, the authors have not clarified some inconsistencies that were (and are) in the classification of the tumors taken into consideration.
For example, lines 106-107 say that 71% of patients (so 84) were in stage T3-4. However, analyzing the numbers of the Cordeiro classification presented in lines 118-120 I find that in 10 patients the defect was type I (limited maxillectomy) and in 49 it was type II (subtotal maxillectomy). If I understand the AJCC-UICC TNM classification correctly, this means that 59 patients (49%) were in stage T1-2. Not only that, in 52 patients the defect was type IIIA/B. So, in how many cases was a “total maxillectomy with preservation of the orbital contents” (Type IIIA) performed and in how many a “total maxillectomy with orbital exenteration” (Type IIIB)? The tumors that require these two types of resection are very different, T3 in the first case and T4a in the second. So, how many patients were stage T3 and how many were stage T4a?
Furthermore, the type of reconstruction is very different if only the orbital floor needs to be reconstructed or if the orbital cavity also needs to be filled and the bed prepared for an ocular prosthesis. Therefore, the lack of these data makes it impossible to understand the exact composition of the cases presented.
Regarding the, in my opinion, incredibly high overall survival and disease-free survival rates, the authors reply: “However, our interpretations show that such a good result could have been achieved because, in addition to very advanced patients, the study included cases with low degrees of malignancy and low local advancement”. So, as I have already pointed out previously, if there was a fair number of patients with “low local advancement” that could justify such high survival rates, how was it possible that 71% of the patients were in stage T3-4?
Furthermore, as many as 40% of cases were N+ in terms of survival. Dozens of studies have shown that paranasal sinus tumors with lymph node metastases have a very poor prognosis. For example, Dalal AJ, et al.1 wrote: “Patients with advanced primary SCC of the hard palate and maxillary alveolus (particularly when there was bone invasion) showed high rates of regional failure, and in most cases successful salvage was not achieved”. Qu Y, et al.2: “With regard to the nodal status, pN0 group had a higher survival rate than pN + group (P < 0.01)”. Kim GE, et al.3: “The overall 5-year survival rate for node-positive patients showed a poorer outcome compared with that for node-negative patients”,
1. Dalal AJ, McLennan AS. Cervical metastases from maxillary squamous cell carcinoma: retrospective analysis and review of the literature. Br J Oral Maxillofac Surg. 2013 Dec;51(8):702-6. doi: 10.1016/j.bjoms.2013.08.011. Epub 2013 Sep 14. PMID: 24041520.
2. Qu Y, Liu Y, Su M, Yang Y, Han Z, Qin L. The strategy on managing cervical lymph nodes of patients with maxillary gingival squamous cell carcinoma. J Craniomaxillofac Surg. 2019 Feb;47(2):300-304. doi: 10.1016/j.jcms.2018.12.008. Epub 2018 Dec 13. PMID: 30595475.
3. Kim GE, Chung EJ, Lim JJ, Keum KC, Lee SW, Cho JH, Lee CG, Choi EC. Clinical significance of neck node metastasis in squamous cell carcinoma of the maxillary antrum. Am J Otolaryngol. 1999 Nov-Dec;20(6):383-90. doi: 10.1016/s0196-0709(99)90078-9. PMID: 10609483.
So I still wonder how it is possible that in a series of patients of which 71% were in stage T3-4 the “Actuarial 5-year overall survival was 113 (95%) and 92 (77%) of disease-free survival (DFS) respectively” (lines 117-118).
Author Response
Comment 1:
For example, lines 106-107 say that 71% of patients (so 84) were in stage T3-4. However, analyzing the numbers of the Cordeiro classification presented in lines 118-120 I find that in 10 patients the defect was type I (limited maxillectomy) and in 49 it was type II (subtotal maxillectomy). If I understand the AJCC-UICC TNM classification correctly, this means that 59 patients (49%) were in stage T1-2. Not only that, in 52 patients the defect was type IIIA/B. So, in how many cases was a “total maxillectomy with preservation of the orbital contents” (Type IIIA) performed and in how many a “total maxillectomy with orbital exenteration” (Type IIIB)? The tumors that require these two types of resection are very different, T3 in the first case and T4a in the second. So, how many patients were stage T3 and how many were stage T4a?
Furthermore, the type of reconstruction is very different if only the orbital floor needs to be reconstructed or if the orbital cavity also needs to be filled and the bed prepared for an ocular prosthesis. Therefore, the lack of these data makes it impossible to understand the exact composition of the cases presented.
Response 1: Thank you for pointing it out. In fact there is a huge difference between reconstructing maxilla with and without exenteration of the orbital cavity. In our observation patients who are well informed about their disease and are aware of the involvement of the orbit (with or without existing unilateral blindness) have much lower expectations regarding the aesthetic effect of the reconstructive procedure. Thanks to your observation the following fragment was added in lines 120-124 "There were 21 patients who underwent a Cordeiro IIIB resection, i.e. total maxillectomy with orbital exenteration (18%). It is worth emphasizing that due to the removal of the orbital content, this group of patients requires a completely different approach in terms of reconstructive techniques and psychological care after surgical treatment"
Comment 2:
Regarding the, in my opinion, incredibly high overall survival and disease-free survival rates, the authors reply: “However, our interpretations show that such a good result could have been achieved because, in addition to very advanced patients, the study included cases with low degrees of malignancy and low local advancement”. So, as I have already pointed out previously, if there was a fair number of patients with “low local advancement” that could justify such high survival rates, how was it possible that 71% of the patients were in stage T3-4?
Furthermore, as many as 40% of cases were N+ in terms of survival. Dozens of studies have shown that paranasal sinus tumors with lymph node metastases have a very poor prognosis. For example, Dalal AJ, et al.1 wrote: “Patients with advanced primary SCC of the hard palate and maxillary alveolus (particularly when there was bone invasion) showed high rates of regional failure, and in most cases successful salvage was not achieved”. Qu Y, et al.2: “With regard to the nodal status, pN0 group had a higher survival rate than pN + group (P < 0.01)”. Kim GE, et al.3: “The overall 5-year survival rate for node-positive patients showed a poorer outcome compared with that for node-negative patients”,
Response 2:
I completely agree with the opinion that advanced cancers originating from the paranasal sinuses have a very poor prognosis. However, in our study, in a large proportion of patients, squamous cell carcinoma most likely (in many cases we were unable to determine the site of the tumor's exit) originated from the area of the oral mucosa of the alveolar or palatine process. Based on the AJCC-UICC TNM classification, each Tumor invades through cortical bone is classified as T4. Considering the fact that patients who underwent any maxillectomy were included in the study, regardless of, for example, the size of the tumor, most of them had the T4 feature. In the case of patients where we were certain that the tumor originated from the maxillary sinus, in less advanced patients, the T4 feature did not occur as often. However, I would like to emphasize that in the case of tumors of the middle face, due to their characteristic anatomy, even smaller tumors can simultaneously invade 3 or more structures and determining the primary exit, as emphasized in the manuscript, is practically impossible. This is undoubtedly one of the greatest limitations of our study.
In the case of the so-called N+ feature. As we well know, nodal metastases in the middle face occur much less frequently than in the case of, for example, the lower face. In our center, if during qualification for the procedure, suspicious lymph nodes are found in the imaging examination, patients are qualified for elective cervical lymphadenectomy in levels I-III, and sometimes even (in the case of skin lesions) with additional removal of the superficial lobe of the parotid gland. In our study, metastatic lymph nodes were found in some of these patients in the above locations. We believe that such a procedure allowed us to obtain such survival results.
Once again, thank you very much for the time you devoted to a thorough analysis of my work. Your advice and the limitations you have rightly highlighted will allow us to design a multi-center study in the future, in which the site of tumor origin will be determined much more precisely. Some of the patients included in our study were diagnosed and treated before 2005, where the preoperative CT scan descriptions were not as accurate as at the end of our study. This may be another limitation.
Reviewer 3 Report
Comments and Suggestions for Authors
I believe that the current form of the article corresponds to the standards of Cancers Journal. The article can be published.
Author Response
Thank you very much for your review and interesting observations.
Round 3
Reviewer 2 Report
Comments and Suggestions for Authors
With this second response from the authors I was finally able to understand the real composition of the presented series of patients, the most frequent site of the tumor, its extension, and the incredibly high overall survival and disease-free survival rates for T3-4 tumors.
Finally, the authors clarified that the great majority of tumors were located in the hard palate-upper gum with small extension into the maxillary sinus, although they invaded other structures. The authors admit that “in a large proportion of patients, squamous cell carcinoma most likely … originated from the area of the oral mucosa of the alveolar or palatine process”. I think these clarifications should be added to the manuscript.
I finally also learn that the authors staged the tumors with the oral TNM classification and not with the maxillary one. But this fact was not (and is not) specified in the manuscript. Since the title already speaks of “Advanced Mid-face Cancer” the reader naturally thinks of tumors of the paranasal sinuses and, therefore, of the TNM classification of these tumors. In fact, in my first comment to the authors I had assumed that the maxillary classification had been used, citing it. Inevitably, therefore, I was surprised by the high survival rate of patients with T3-4 tumors. In fact, one of the biases of the TNM classification is that the same carcinoma of the hard palate-upper gum with small invasion of the maxillary sinus is classified T2 with the maxillary classification and T4 with the oral cavity classification. So, a T3-4 tumor of the upper oral cavity is very different from a T3-4 tumor of the maxillary sinus.
So, in my opinion, the authors must specify on lines 105-107, or elsewhere, that the tumors were staged as oral cavity tumors.
After this clarification, the high percentage of lymph node metastases also finds its explanation. A series of studies have, in fact, demonstrated that maxillary tumors involving the hard palate and upper gum metastasize regionally more than those intrinsic to the maxilla. For example, Montes et al. say: “oral cavity maxillary squamous carcinomas have a cervical metastatic rate that is higher than expected and approaches that of tongue and floor of mouth”.1
And Zhang et al., in their review and meta-analysis on this topic, wrote verbatim: “Based on our literature review and meta-analysis, we conclude that the risk of cervical metastases for SCC originating from the maxillary gingiva and the hard palate is higher than expected and is comparable to that of other oral sites”.2
A brief explanation of these data (perhaps with citation of the studies I indicated or of the numerous others on this topic) should be added to line 108 where it says “…regional lymph nodes (39%)…”.
1. Montes DM, Carlson ER, Fernandes R, Ghali GE, Lubek J, Ord R, Bell B, Dierks E, Schmidt BL. Oral maxillary squamous carcinoma: an indication for neck dissection in the clinically negative neck. Head Neck. 2011 Nov;33(11):1581-5. doi: 10.1002/hed.21631. Epub 2010 Dec 6. PMID: 21990223.
2. Zhang WB, Peng X. Cervical metastases of oral maxillary squamous cell carcinoma: A systematic review and meta-analysis. Head Neck. 2016 Apr;38 Suppl 1:E2335-42. doi: 10.1002/hed.24274. Epub 2016 Feb 18. PMID: 26890607.

Author Response
Response : In accordance with the recommendations, all recommendations indicated by the reviewer have been added in lines 112-124. This concerns the indication of a specific TNM classification, presentation of the theory of the starting point of tumors and discussion of nodal metastases together with the addition of citations indicated by the reviewer. Once again, I thank you very much for the thorough analysis of the work. The corrections made have improved the substantive value of the article immensely.